# Insect Production for Animal Feed: A Multiple Case Study in Brazil

**Jaqueline Geisa Cunha Gomes** [1,*], **Marcelo Tsuguio Okano** [1,2], **Edson Luiz Ursini** [2] **and Henry de Castro Lobo dos Santos** [2]

1 UNIP—Graduate Program in Production Engineering, Universidade Paulista, São Paulo 04026-002, Brazil; marcelo.okano@unip.br
2 UNICAMP—College of Technology, State University of Campinas, Limeira 13484-332, Brazil; ursini2@unicamp.br (E.L.U.); h190839@dac.unicamp.br (H.d.C.L.d.S.)
* Correspondence: jaqueline.gomes34@aluno.unip.br

**Abstract:** The production of insects as a sustainable protein source represents an innovation for animal feed. The objective of this research is to analyze the value chain of the use of edible insects in animal feed in Brazil through the framework of SWOT, the business model sustainable canvas, and a multiple case study, highlighting the sustainability characteristics. A qualitative approach of the descriptive exploratory type was used, and the multiple case study identified the actors in the chain and how value is generated. The young age of the sector explains the characteristics observed in the Brazilian chain, such as a large development deficit in terms of financing, technology and the qualification of human resources; a disorganized supply chain and supplier structure; and efforts undertaken by regulatory agencies to promote the development of regulations relating to the production and use of insects in animal feed, which, in turn, will lead those wishing to participate in this innovative venture into research and development in the area. Brazil's edible insect supply chain can become a more significant aspect of sustainable agriculture by closing nutrient and energy loops, promoting food security and minimizing climate change and biodiversity losses, all of which are associated with the achievement of the Sustainable Development Goals.

**Keywords:** value chain; edible insects; insect cultivation; animal nutrition

## 1. Introduction

The global demand for feed, as well as the competition for protein, grows annually. Research is carried out worldwide to mitigate possible protein production shortages, which will need to be increased by sixty percent by 2050 to meet future world demand [1]. Another aspect affecting the animal protein production system is the high inputs required to produce feed, the global demand for which will reach more than 1 billion tons by 2050 (an increase of 60 to 70% compared to around 800 million tons in 2018) [2].

One of the biggest challenges related to this future demand is increasing the availability and simultaneously reducing the use of natural resources, such as land for soybean planting and water, as well as reducing greenhouse gas emissions. The use of insects as a partial or total ingredient in animal feed has been shown to represent an alternative source of protein and a substitute for traditional feed.

Some research has demonstrated that the consumption of red meat is associated with an increased probability of stroke, diabetes, colon cancer and lung cancer. Insects seem to have a more nutritious and healthier composition than meat-based foods, and they are also diverse in terms of nutritional value. As a result, they can be used as meat substitutes [3,4]. These problems encourage a decrease in meat consumption and its replacement with insect consumption. This is valid not only because they offer replacement protein, but because of insects being a more sustainable, healthy and economical product.

Studies such as those performed by Chia et al. [5] show that animal protein producers are aware of the potential related to using insects as a feed ingredient, and that producers' knowledge is directly proportional to their willingness to pay for this new type of feed.

In the study "Insects as a sustainable feed ingredient in pig and poultry diets—a feasibility study" [6], the actors involved in the production chain of using insects as a sustainable ingredient of pork and poultry feeds are demonstrated, along with a discussion of how their integration can influence the implementation of insects as an alternative source.

This article aims to analyze the value chain of using edible insects in animal feed in Brazil through the framework of SWOT and a sustainable business model canvas and multiple case study, highlighting the sustainable characteristics. As this value chain is still in its nascent stages in Brazil, the following actors were not considered: the poultry/pork sector and retail/consumers.

## 2. State of the Art

The most promising insects for use in industrial feed production are the black soldier fly (BSF) (*Hermetica Illucens*), the housefly (*Musca domestica*) and the yellow mealworm (*Tenebrio molitor*) [6].

Zhou et al. [7] summarized the nutritional value of different classifications of edible insects. The composition of the nutritional value of edible insects is illustrated in Figure 1.

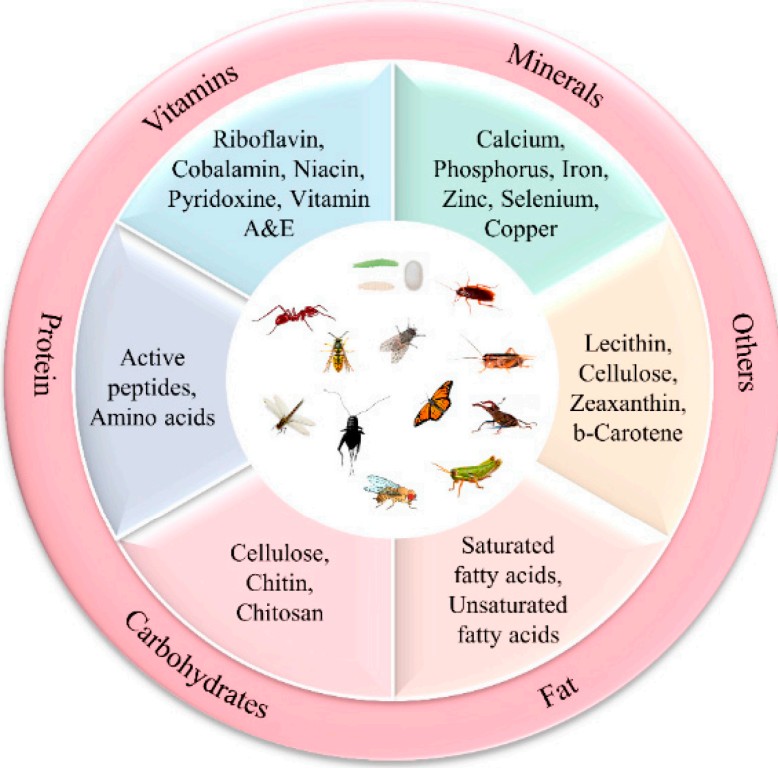

**Figure 1.** Nutrient composition of edible insects. Based on Zhou et al. [7].

The general composition of edible insects is listed in Figure 1. Different units are employed to represent the data due to the different sources from which they are derived, and the lack of their completeness is a result of the conversion units used. The material composition of insects is markedly different between species. In the dry matter, protein and fat are the most common substances [7].

Insects have tremendous potential at all life cycle stages as sources of nutritional value; they are a significant source of animal protein, contain essential amino acids and minerals (K, Na, Ca, Cu, Fe and Zn), and their fatty acids are unsaturated. The assimilation rate of

insect proteins is 76–82%. Insect carbohydrates are primarily composed of chitin, which is present at a concentration range of 2.7 mg to 49.8 mg/kg of dry mass [8].

Table 1 presents Shah et al.'s [9] assessment of the types of economical insects and their chemical compositions and nutritive values.

**Table 1.** Types of economical insect and their chemical composition and nutritive value.

| Insects Species | Percentage (%) | | | | | | | | | Milligram per Kilogram (mg/kg) | | | | |
|---|---|---|---|---|---|---|---|---|---|---|---|---|---|---|
| | DM | CP | CF | Ash | Ca | P | Mg | K | Na | S | Zn | Cu | Mn | Fe |
| Black soldier fly larvae | 27.40 | 56.10 | 23.20 | 9.85 | 2.14 | 1.15 | 0.39 | 1.35 | 0.13 | 27.04 | 13.10 | 11.20 | 23.20 | 20.40 |
| Housefly larvae | 83.47 | 33.29 | 6.20 | 6.25 | 0.49 | 1.09 | 0.23 | 1.27 | 0.54 | ND | 10.39 | 32.40 | 42.50 | 47.50 |
| Mealworm larvae | 94.60 | 55.83 | 25.19 | 4.84 | 0.21 | 1.06 | 0.30 | 1.12 | 0.21 | ND | 138.2 | 29.40 | 05.70 | 71.50 |

DM, dry matter; CP, crude protein; CF, crude fiber; Ash; Ca, calcium; P, phosphorous; Mg, magnesium; K, potassium; Na, sodium; S, sulfur; Zn, zinc; Cu, copper; Mn, manganese; Fe, iron. Source: Shah et al. [9].

The DM content in fresh BSFL (Table 1) is greater than that of other products (34.9 to 44.9%), which results in BSFL being more affordable and easier to make. Typically, BSFL has a composition of 41.1 to 43.6% CP, 15.0 to 34.8% EE, 7.0 to 10% CF, 14.6 to 28.4% ash, and 5278.49 kcal/kg GE, based on DM [10,11]. BSFL larvae are high in Ca (5 to 8%) and P (0.6 to 1.5%). Additionally, their mineral composition contains Cu (6.0 mg/kg), Fe (0.14–14%), Mn (246 mg/kg), Mg (0.39), Na (0.13), K (0.69%) and Zn (108 mg/kg) [9,12,13].

On average, housefly larvae contain 6.25% ash, 83% DM, 33.29% CP, 6.2% CF, 0.49% Ca, 1.09% P, 0.23% 1.27% Mg, 10.39 mg/kg Zn, 32.40 mg/kg Cu, 42.50 mg/kg Mn and 47.50 mg/kg Fe, based on DM (Table 1) [9]. Housefly larvae contain a lot of energy, protein and micronutrients (e.g., Cu, Fe and Zn), as well as EAA and FA. They are also inexpensive, have high nutritional value and are easier to access than other sources of animal protein [9].

According to Shah et al. [9], the average total DM composition of mealworms is 94.6%, comprising CP 55.83%, CF 25.19%, ash 4.84%, calcium 0.21%, phosphorous 1.06%, Mg 0.3%, K 1.12%, Na 0.21%, Zn 138.2 mg/kg, Cu 19.4 mg/kg, Mn 5.7 mg/kg and Fe 71.50 mg/kg (Table 1).

The edible insect sector has attracted global attention, and this has led to a re-assessment of the practice of entomophagy, or the consumption of insects, in countries that typically show reservations, as well as in countries that are entering the developmental stages of the practice [14].

In Europe, the aquatic insect feed market constitutes approximately half of the entire animal-based insect feed market. Its growth is expected to reach 75% in the next 6 years, and European breeders of insects currently carry around 1000 tons of protein-based insect feeds [15]. Since 2021, in the European Union and member states, processed animal proteins (PAPs) have been permitted for use in poultry and swine feed. This approval represents one of the first steps in the general authorization process, and will ultimately guarantee long-term resource utilization in animal protein production, thus addressing environmental concerns [15,16].

According to Lähteenmäki-Uutela et al. [16], in many parts of North America, insects have historically been incorporated into the food culture. The farming of insects intended for food and feed production began to increase following 2012. The modern insect industry comprises companies that already cultivate crickets and mealworms for animal food. To avoid the high costs of labor, many American and Canadian insect farms have invested in robots, automation, sensors and data aggregation.

Many countries in Africa have insufficient or no insect-specific laws, regulations, standards, or labeling in place to regulate the production and distribution of insects in

food or feed chains. The absence of a solid foundation is a significant obstacle to the establishment of markets for insects and their related products [17]. There is a need for more enhanced technology in the rearing of insects to address the increasing pressures of population growth, as we can no longer rely on the catching of wild insects [17].

In several Asian countries, insects have historically been regarded as a form of food, and been used as a significant source of protein. In China, there are no specific laws that regulate their production. Other insects can also be utilized as food additives, and in this context, producers must follow the regulations established in the Administrative Measures for Food and Feed Additives [18]. In South Korea, insects have been a part of the human diet for centuries; they are also included in animal feeds, and there are no specific rules restricting the food and feed industry in relation to insects due to a legislative liberalization that took place in 2015 [19].

In Brazil, the food insect chain is still in its early stages of development, and there is still a significant lack of development in terms of funding, technology and the qualification of human resources. There have also been efforts made by regulatory agencies to promote the development of regulations pertaining to the production and utilization of insects in animal feeds, which will, in turn, encourage those who want to participate in this innovative endeavor [20].

Currently, in Europe, there is a supply–demand gap in the edible insect chain [15]. This scenario also applies to Brazil, as agrofactories still cannot access the necessary volumes (of consistent quality) required for processing insects, and the feeds produced are not competitive in terms of their costs compared to conventional protein sources. In the Brazilian scenario, logistical issues make the process more expensive, and hinder the growth of the chain. Furthermore, there is no knowledge on the part of protein producers about the effects of insects as a feed ingredient.

Another major challenge is that induvial companies are involved from the beginning to the end of the process. As such, they represent a significant part of the chain, from insect breeding to processing, thus raising production costs. As has been seen in Europe [15], it is expected that in the future, Brazilian companies will specialize as the value chain matures.

Shah et al. [9] have shown that a hypothetical expansion of commercial farms (that spend EUR 1000 per month on SBM-derived protein) would involve completely replacing SBM with BSF, HF or MW. The extra costs associated with these species are EUR 88,230, 3980 and 13,010, respectively. If we consider that the farmers farm every season, it would be extremely expensive for them to turn over the production of a whole field. In order for insects to be considered a viable alternative to SBM and FM, both cost and nutritional value must be maintained. For this to be successful, expenditure on insects must be reduced to EUR 0.4 per kilogram of direct weight based on 35% DM substances [9]. The economic values of different insects compared to other sources of protein are shown in Table 2.

A very important element of the end of the chain is the acceptance of insect-based feeds. The study carried out by Ankamah-Yeboah et al. [21] showed that 77% of the people interviewed are indifferent regarding the use of insects in animal feed for fish, which is a promising result for the aquaculture industry.

Not all insects can only be used as ingredients in animal feed. According to Čičková et al. [22], some insects can play dual roles within the chain, such as via the recycling of their organic by-products in compost fertilizers, and the use of their protein as feed. Another challenge encountered at the beginning of the chain relates to the food that insects can consume. These include residues, such as perishable organic by-products. These by-products are a valuable energy source for insects, but require producers to employ preservation techniques that both maintain the nutritional quality of the biomass and are economically viable [23].

In 2022, each person in Brazil generated an average of 1.043 kg of waste per day; that is, almost 1 kg per person. In general, 81.8 million tons of domestic waste were produced, which corresponds to 224 thousand tons per day (2022 edition of the *Panorama of Solid Waste in Brazil*, by ABRELPE) [24].

**Table 2.** The economic value of insects compared to other protein sources.

| Potential Source | Housefly Maggot | Black Soldier Fly | Mealworm | Fishmeal | Soybean Meal |
|---|---|---|---|---|---|
| CP (%) | 50.4 | 42.1 | 52.8 | 75.4 | 52.00 |
| Lysine (%) | 6.1 | 6.6 | 5.4 | 7.5 | 6.3 |
| Methionine (%) | 2.2 | 2.1 | 1.5 | 2.8 | 1.3 |
| PPR (EUR/kg) | 1.08 | 20 | 3.7 | 1.24 | 0.2 |
| PP (EUR/kg) | 2.14 | 47.51 | 7.01 | 1.64 | 0.54 |
| PL (EUR/kg) | 0.13 | 3.14 | 0.38 | 0.12 | 0.03 |
| PM (EUR/kg) | 0.05 | 1.00 | 0.11 | 0.05 | 0.01 |
| PP TO PP SBM [1] | 3.98 | 88.23 | 13.01 | 3.05 | 1.00 |
| PL TO PL SBM [1] | 3.85 | 92.23 | 11.15 | 3.64 | 1.00 |
| PM TO PM SBM [1] | 6.73 | 142.52 | 15.02 | 6.58 | 1.00 |

CP, crude protein; PPR, product price; PP, protein price; PL, price of lysine; PM, price of methionine; PB, protix biosystems; AP, agriprotein. [1] PP to PP SBM, price of replacing 1 kg of protein derived from SBM with other protein sources; PL to PL SBM, cost of replacing 1 kg of lysine derived from SBM with lysine from other protein sources; PM to PM SBM, cost of replacing 1 kg of methionine from SBM with methionine derived from other protein sources. Source: Shah et al. [9].

In Brazil, organic waste represents approximately 50% of all solid waste generated. Dry recyclables (28%) and waste (22%) are the next most prolific elements. The size of the organic fraction affirms the importance of using these residues in different ways, and avoiding their unnecessary disposal [25]. In addition, less than 2% of organic waste is currently composted in Brazil [25], and the storage of large batches of waste is a problem, since it cannot be immediately offered to insects [23].

Black soldier flies (BSFs) have received increased attention in recent years because of their capacity to contribute to sustainable waste management and renewable energy. In Brazil, the species that currently receives the most attention is the BSF. BSF larvae are voracious feeders and have the capacity to convert food waste into protein and lipid products [26]. Within the field of sustainability, BSFs are considered an interesting solution to reduce the ecological impacts of food waste. In transforming waste into valuable products, BSFs contribute to reducing the emission of environmental pollutants and greenhouse gases [26]. We highlight below some experiments carried out on BSFs and reported in published scientific articles and Master's theses.

Silva et al. [27] aimed to create a demonstration unit for black soldier fly larvae (*Hermetia illucens*) using organic waste from a restaurant located at the Federal University of South and Southeast Pará. It was known from previous experiments that the production of larvae of the black soldier fly can be used to efficiently decompose organic residues, transforming them into liquid composts and soils ready to be used in the cultivation of vegetables and the preparation of seedlings.

In Santos' Master's thesis [28], the objectives involved valuing the by-products of the food industry, obtaining a potential ingredient for animal feed, and also verifying the influence of diet on the weight, length and nutritional composition of BSF larvae. The results show that the diet significantly influenced the weight and average length of the larvae. Regarding the nutritional composition of the larvae, the protein content was not altered by the diet supplied, but the fat and ash contents of the diet directly influenced the composition of the larvae.

Teixeira Filho [29] proposed a possible solution to mitigate two issues: the disposal of solid organic waste and the pressure on the current supply of food protein. This solution is based on the mass production of larvae of *Hermetia illucens* (L., 1758) (Diptera: Stratiomyidae), also known as the black soldier fly, to degrade organic solid waste and also as an alternative source of animal protein. They achieved an 83.75% reduction in organic solid waste by employing a solid-waste-to-protein biomass conversion rate of 23.2%. This work provides a good indication that the mass production of the black soldier fly to enable the destruction of organic waste and for subsequent use as a source of animal protein is

an excellent and sustainable alternative solution to problems related to solid waste and protein supply.

The treatment of organic waste is related to several SDGs, but specifically to numbers 2 (Zero Hunger and Sustainable Agriculture), 12 (Responsible Consumption and Production) and 13 (Action Against Global Climate Change).

Agricultural and food or agro-food supply chain (AFSC) encompasses all stages, from cultivation to harvesting, packaging, processing, transporting, marketing and distribution and final consumption. It not only entails general risks, including social, political, cultural and economic; due to the perishable nature of the products, seasonality, weather effects and quality and safety requirements, the chains are even more vulnerable [30–32]. The productive chain of edible insects operates like an agro-food chain, and to verify its characteristics, we will employ two frameworks.

All parts of the value chain of producing edible insects as an alternative protein source for animal feed, and the associated challenges, can be analyzed through the sustainable business model canvas, as shown in Figure 2.

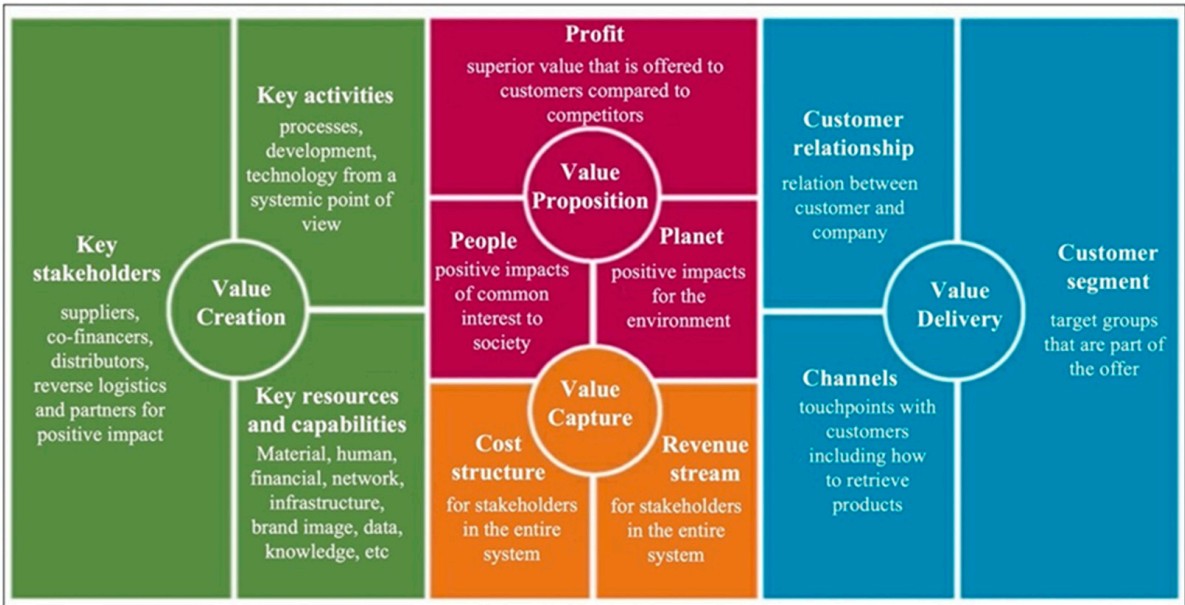

**Figure 2.** Screenshot of the sustainable business model canvas. Source: adapted with permission from Bocken et al. [33], Osterwalder and Pigneur [34] and Richardson [35].

Figure 2 depicts the business model canvas and introduces three new archetypes into the value proposition aspect: profit, people and planet. These make it a sustainable model with a holistic approach, the main (and most challenging) objective of which is to act sustainably in the future, with a simultaneous focus on environmental, economic and social changes [33].

The supply chain of edible insects will be evaluated using the SWOT matrix. SWOT is a helpful tool that can be used during the evaluation phase in order to yield a first interpretation of the possible future consequences. SWOT analysis is a simple method that provides a factual interpretation of the benefits and drawbacks of a business [36].

According to Benzaghta et al. [36], the SWOT matrix (Figure 3) can be summarized as follows:

- SO strategies—taking advantage of opportunities;
- ST strategies—avoiding threats;
- WO strategies—introducing new opportunities by reducing weaknesses;
- WT strategies—avoid threats by minimizing weaknesses.

**Figure 3.** Screenshot of the sustainable business model canvas. Source: Based on Benzaghta et al. [36].

## 3. Methodology

This qualitative research approach considers the current state of the art of using insects for animal feed, organic waste and other sustainable actives. To enact this evaluation, bibliographical research was carried out, using the search strings "value chain" and "edible insects" in the databases Web of Science and Scopus.

To meet the research objective, articles with information on the use of edible insects for animal feed and organic waste were selected. The intention was to present the current state of the art in this field of research and to broadly contextualize it, highlighting the value chain of this agri-food chain in Brazil by focusing on two companies that operate in the area. A descriptive exploratory case study was carried out, and through interviews it was possible to analyze the business models of two Brazilian companies active in the edible insect sector.

For confidentiality reasons, the companies are named Alpha and Beta. Their main characteristics are described below.

Alpha company was created in 2015 with the aim of contributing to the development of sustainable food, focusing on insects with high nutritional quality as a protein source. The company's mission is to work towards resolving the food issue by 2050. The company seeks to have a social and environmental impact, in addition to generating profit. The owner of the company first encountered edible insects in 2013, according to a report by the FAO and a later interview undertaken with the researcher, professor doctor Van Huis.

Beta company was created in 2022 with the aim of ensuring food safety and quality in producing animal feed. The company produces protein and oil through a circular process. The company began operations about three and a half years ago with a specific project, and it has positioned itself in the market as a company that combats hunger, is sustainable and combines financial prosperity with the protection of the environment.

The methodological procedure employed consists of the following steps:

1.  Sources of evidence for the case study—As a source of internal evidence, interviews were conducted with the owners of Alpha and Beta companies. External evidence sources were not used, as the companies do not have websites containing relevant data for this research, and it was not possible to access internal documentation, such as meeting minutes, process reports or quantitative data.
2.  Research instruments—A pilot interview was carried out with one of the companies to assess whether the theoretical concepts reflect the day-to-day activities of the company. The pilot interview aimed to delineate the actors in the chain, including those related to biowaste, the rearing of insects, the processing of insects, the feed sector, the protein sector and retail/consumer. Subsequently, the final interview script was prepared based on the sustainable business model (Figure 1) [33–35];
3.  Data collection—Interviews were conducted using a semi-structured script and the business owners were interviewed. These were scheduled and performed remotely, and were recorded and transcribed. Each interview was approached as a detailed case study of the parts of the respective business;
4.  Data results—The collected data were compared with the theoretical assumptions. The interviews were carried out to understand the effectiveness of the actors in the chain, as well as the maturity of the companies and the difficulties encountered by entrepreneurs in the edible insect sector in Brazil;

5. Discussion and conclusion—The case studies yielded a practical understanding of the social, economic and environmental conditions of these companies. They elucidated the market transition that one company is currently undertaking, as well as the market positioning that both hope to achieve.

## 4. Results

### 4.1. Brazilian Edible Insects Supply Chain

The SWOT matrix was applied to analyze the Brazilian edible insects supply chain. This approach highlights the following:

Strengths:

- Insect farming generates less greenhouse gas than traditionally farmed cattle;
- The reproduction of insects using food waste also facilitates the dissipation of large amounts of organic waste;
- Reduced requirement for fresh fruit and grain flour (as currently used in insect breeding);
- The production of insect protein as a replacement for animal protein will reduce the consumption of red meat, and is more sustainable, healthy and economical;
- Insect farming requires less water than producing the same amount of animal protein.

Weaknesses:

- Legislative bans on insects and insect-based products that are intended for commercialization as food;
- Artificial diets based on food byproducts should be specifically studied for each species of edible insect;
- Only five companies currently work with edible insects.

Opportunities:

- Insects have a promising history of being used to produce proteins and fat, and they serve as an effective source of these two substances, thus helping to combat protein energy deficiency while minimizing the environmental impact of food production.

Threats:

- The presence of substances derived from organic materials (e.g., herbs) that are potentially harmful to insects;
- Cultural impediments to the introduction of edible insects into animal feed;
- The potential for heavy metals and mycotoxins to be bioaccumulated;
- The mechanisms of pesticide, drug and hormonal uptake are uncertain.

### 4.2. Characteristics of the Companies Studied

The sustainable business model canvas was used as a model for the analysis of companies. In addition to the nine blocks of the traditional business model canvas (customer segment, customer relationship, channels, cost structure, revenue stream, key activities, key resources and capabilities, key stakeholders and value proposition), this model incorporates the 3Ps of value proposition related to sustainability:

- Profit—superior value that is offered to customers compared to competitors;
- People—positive impacts of common interest to society;
- Planet—positive impacts for the environment.

Via the two case studies, it was possible to identify the actors in the chain, their roles, difficulties, and how value is generated.

#### 4.2.1. Value Proposition

Alpha Company

- Profit: The company's value proposition relates to the production of sustainable feed, aiming at preserving the environment, as well as aiding the economic development of several Brazilian communities. The company's ambition is to create mechanisms that can prolong life and restore Brazilian ecosystems. At present, the development

in this company of insect processing for animal feed relies on a single pilot plant, but one with high capacity and the potential to become the largest insect processing plant in Brazil and Latin America. The plant will begin serving the aquaculture, poultry and pet sectors of the animal protein market. Wirth a vision of future markets, the company invests in research into the use of insects in the pharmaceutical sector and has an expansive growth strategy extending up to 2027.

- People: The company's strategy is to open up the production process and the technology used in production, contributing to the expansion of the production chain, and consequently to increase employment. To this end, the company will supply insect eggs through partnerships, and will share their cultivation methods with new breeders who will become part of the chain. In this way, it will be possible to impart better living conditions upon small producers, offering these families a better income and a better quality of life.
- Planet: The insect used to produce feed is BSF, which is fed on waste. Organic waste, for example, that was previously sent to landfills, is now reintroduced into a new chain as food. However, the company faces challenges in accessing this waste. Some of these challenges include the cost of transporting the waste to the destination, the collection of large volumes in different locations and the duration for which the waste can be deposited while awaiting collection. Other types of waste can be used, such as brewery waste. However, this waste is about seventy percent water, which makes freight and transport extremely expensive, increasing operating costs. Depending on the type of waste, the logistical strategy changes, as the requirements of waste removal are inconsistent. In addition, the requirements when storing waste in its place of origin also differ from those of its place of origin. To overcome the challenges of waste collection, the company visits market actors involved in composting and landfills, and it has discussions with managers in the urban and organic waste chain. Once these challenges are overcome, the use of waste can contribute to mitigating environmental problems.

Beta Company

- Profit: The company's value proposal is to extinguish hunger and accelerate the ecological regeneration of Brazil, via the production of high-quality protein and oil, thus combining financial profit and sustainability. The company is now exiting the laboratory stage and looking to build a pilot plant for growing and processing BSF, which will be registered with the Ministry of Agriculture. The company's ambition is to become the largest animal feed production company in Latin America. The company enacts the entire process, from the collection of organic waste to the production of flour and insect-based oil used for animal consumption. In addition, the company markets the eggs it produces. As such, the company is involved in the collection of urban waste, in the transformation of this organic residue into substrate, in the fattening of the larvae and in the processing of the larvae for product formulation.
- People: Via the decompression of the protein chain, the company can work towards the fight against hunger by reducing the competition between protein sources for animal feed. When insects are introduced as an integral component of animal feed, it will be possible to guarantee a better supply of protein for people. Furthermore, greater understanding will lead to behavioral changes that will contribute to responsible consumption.
- Planet: At the company's center of operations, approximately three hundred and sixty-six tons of urban organic waste are generated per hour. The work of transforming this waste into substrate for the BSFs further significantly impacts daily waste production, even if to a smaller degree. In this way, organic waste is transformed into yeast for the development of BSF larvae, which are the raw materials for dog, cat and ornamental fish feed. As these animals' diets compete with human diets over the same protein sources, the proposal is to reduce the pressure on the protein production chain, enabling a reduction in the use of natural resources.

4.2.2. Value Creation

Alpha Company

- Key stakeholders: The company is partnered with universities for biological and engineering research. This partnership is based on a win–win theory. As a result of this collaboration, the company incurs no costs in laboratory research. In addition, the company's interns use the universities' laboratories for research. In the production of BSFs, the company is partnered with another company that specializes in organic protein production. In addition, the company is also partnered with the Brazilian Micro and Small Business Support Service (SEBRAE), which facilitates national and international fairs, increasing the company's visibility.
- Key activities: As the BSF processing methodology is not yet perfected, the company offers insects to other companies that produce biological agents. Further, they encourage the use of insects in different stages of their life cycle as food—including pulps, young insects and adults. The company undertakes the entire rearing process, separates, weighs and packs them, issues invoices and dispatches the insects to the end customer.
- Key resources and capabilities: The company's greatest resource is the formula with which they feed the insects, which comprises wheat bran, cornmeal and corn, in addition to hydrated vegetables. Another important resource is knowledge of the parameters of the breeding process, such as temperature and humidity. This knowledge was originally acquired experimentally, and continues to be improved since the ideal production model remains unknown.

Beta Company

- Key stakeholders: The main business partner is in the organic food composting sector, and this company is located in the same place as Beta company. Thus, the logistical costs are very low. This partner is also a co-founder of Beta company. The company also has partnerships with rural federal universities.
- Key activities: The company currently produces fresh insects to meet animal production needs.
- Key resources and capabilities: The company's main resources are the equipment and physical space that comprise the factory, in addition to labor.

4.2.3. Delivery of Value

Alpha Company

- Customer segment: The company has three customer segments. The first segment is customers who buy live insects and resell them for animal feed. The second segment is the final customers, who buy the insects to feed their pets. The third segment is customers who use the insects as biological agents.
- Relationship with customers: As the market is small, the relationship with customers is close. Due to its origins as a family business and its few employees, customer needs are addressed promptly.
- Channels: The most commonly used communication channels are social networks and WhatsApp. The company has a website that is currently being restructured, as the company's business model is undergoing development.

Beta Company

- Customer segment: When the company's plant is up and running, the main customer segment will be feed producers. Subsequently, the company will have its own feed line, mainly for domestic animals.
- Relationship with customers: The relationship with customers will be developed as the company matures. However, the company is already positioned in the market, and sells live insects.

- Channels: The company has a structured website, through which it is possible to contact them and ask questions about the products offered. It is also possible to contact the company by email and LinkedIn.

### 4.2.4. Value Capture

Alpha Company

- Revenue stream: All three market segments offer similar revenues. However, the revenue source offering the highest margin is the segment of final customers who buy insects to feed their non-conventional pets. This market is seasonal, depending on the type of animal; for example, frogs are fed live insects only in parts of their life cycle.
- Cost structure: The company's biggest costs are raw materials and labor. Under the new model of producing insect-based feeds, the biggest cost will be the processing of the feed.

Beta Company

- Revenue stream: The company's revenue is negligible, as it is still under development. However, the company predicts that 80% of its revenue will come from insect flour.
- Cost structure: The highest costs are related to the construction of the factory and the purchasing of equipment for BSF production, as well as electricity and general maintenance.

## 5. Discussion

Brazil's edible insect supply chain will play a significant role in circular sustainable agriculture via its capacity for closing nutrient and energy loops, promoting food security and minimizing climate change and biodiversity loss. These aspects strengthen the chain and are associated with the achievement of the Sustainable Development Goals.

The network is still young and disorganized, as only five companies are in operation and there are some factors that contribute to slowing development, such as legal obstacles against and a lack of legislation for insect-based products, and the lack of cultural acceptance of the consumption of insects in Brazil, as verified in the weaknesses section and pointed out by others [20]. Another characteristic of this chain is that the companies work independently and autonomously; that is, each performs all parts of the production process, and there is no network of collaboration among them.

The case studies presented elucidate certain as-yet-unmet opportunities to contribute to the sustainable development of the chain, such as the use of land for soybean planting; reducing water consumption and greenhouse gas emissions; replacing animal proteins, and reducing organic material, among others.

The sustainable business model canvas enabled us to analyze how both companies are structured. Despite only focusing on two case studies, the conditions of these companies reflect the contemporary condition of the value chain associated with the production of edible insects for animal feed in Brazil. As such, we can conclude that the chain is still young and that its development will be complex, since the production process is still under development.

As regards the business models, we can observe adhesions between practice and theory. The following constitute the pillars of the business model.

Value proposition: The companies' value propositions reflect concerns about the planet, and especially about the future demand for protein and the competition that will arise between food and feed, as well as the demand for fishmeal for fisheries [5]. The desire to preserve the environment has encouraged businessmen to seek new solutions in structuring the animal feed market. However, international research on the protein capacity of insects, as well as millionaire-funded projects, have been decisive in encouraging investment in a new market, which features sustainability as its main motivation.

Similarly to Europe [15], in Brazil, insect-based protein products still cannot compete with established protein sources, in terms of costs. Further, agrofactories still do not have the capacities or consistency in quality required for processing.

As identified by [22], some insects can play dual roles within the chain, such as via the recycling of organic by-products into compost fertilizers, and this is also the case for BSFs. BSFs consume waste as a source of energy, and this offers numerous benefits to a developing country that is facing difficulties in the collection of urban waste and has little infrastructure for selective collection.

Furthermore, as described in [6], the chain of using insects as animal feed in Brazil begins with bio-waste, but currently ends with insect processing, given that both the active companies still do not produce processed feed for poultry, pork, or fish. The next steps will be to develop the poultry, fish, pork and pet (i.e., dogs and cats) markets via feeding with processed feed.

Despite the entrepreneurs' willingness to reveal their secrets related to the cultivation of insects, certain aspects related to processing remain concealed, as well as the technology used. Both companies are currently undertaking test phases, and their industrial plants are still in the pilot phase. As such, it is not possible to guarantee, despite the energy expended on research, that their models will work. To arrive at the most optimal model, more tests and improvements will be required until the whole process is matured.

The creation of value in companies is based on partnerships. The chain is developed through strategic partnerships. However, there are differences in the types of partnerships. To produce BSFs, Alpha company is partnered with another that specializes in organic protein, and this company invests in technology and capital; on the other hand, Beta company's main partner is an organic food composting company. In addition, Beta company has gone public, and thus receives investments from private individuals.

Despite BSFs showing promise and representing a good investment for Brazilian businessmen, there is a logistical bottleneck associated with the cost of feeding them, and this represents one of the biggest challenges for the chain. Beta company encounters fewer difficulties due to its partnership, while Alpha company must search for waste depository locations and logistical strategies, even though there are several sanitary landfills in Brazil, as shown in Figure 4.

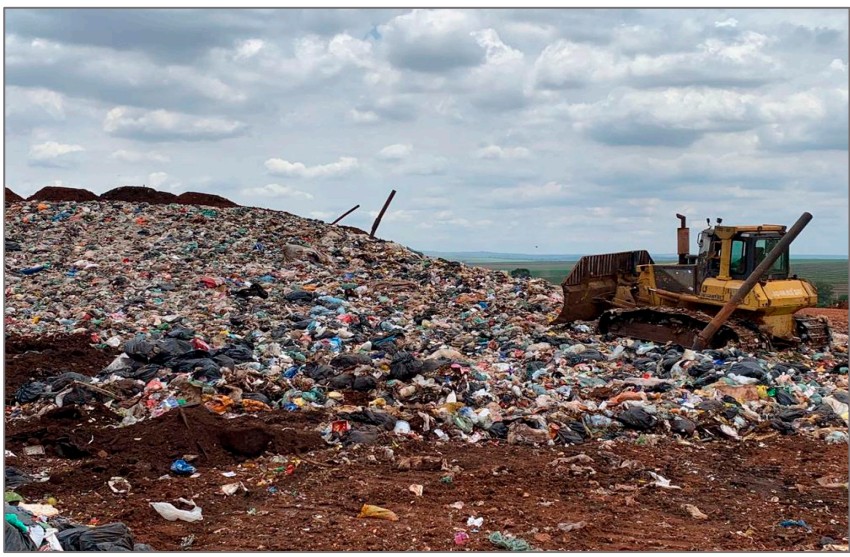

**Figure 4.** Guatapará landfill in São José do Rio Preto, São Paulo. Source: personal field research archive.

## 6. Conclusions

Our analysis of Brazilian companies active in the production of insects for animal feed in Brazil has allowed us to understand the current organizational condition of these companies. Just as international research on the subject is underdeveloped, the companies involved are also young, with Alpha being the most experienced in the sector, but having only been on the market for seven years.

The young age of the sector explains the characteristics observed in the Brazilian chain, such as the high degree of underdevelopment in terms of financing, technology and the qualification of human resources; the disorganized supply chain and supplier structure; the efforts being made by regulatory agencies to develop regulations relating to the production and use of insects in animal feed, which, in turn, should encourage those interested in this innovative venture into further research and development.

As shown in previous surveys undertaken in other countries and in Brazil, for BSFs to be integrated as an animal feed that can reduce organic waste in a sustainable way, companies must invest time and financial resources into research focusing mainly on BSFs, according to the characteristics of this species. However, despite these investments, in Brazil, as well as in Europe, insect-protein-based products are still not competitive compared to established protein sources in terms of cost. The non-competitiveness of the Brazilian chain makes it difficult for it to develop a market that is regulated in terms of prices, products, supply and demand.

Because the productive chain is still in its nascent stages, several aspects of the business model are still being developed, as well as the teams, processes, labor, technology and space required. Further, more time will be required to allow the maturation of the blocks that make up value delivery and capture, further characterizing the difficulties encountered in the Brazilian chain in establishing itself and becoming competitive.

The companies' strategies for dealing with this process were not revealed. However, Alpha company seems to be employing a "homemade" strategy, founded on the time it has spent in the market and its acquired experience. Beta company, on the other hand, seems to believe that investing in advanced technology is the safest path. It will soon be clear which of these strategies is best.

The concepts of sustainability and environmental regeneration increase the value of the benefits offered by companies. However, both these companies desire a market that will grow annually, which is why they are currently racing to be the first to establish processed BSF for use in animal feed. The conditions in Brazil are optimal for the growth and production of BSF, as a large volume of organic waste is generated daily and the species is native to this country.

Previous research and our study of the Brazilian edible insect production chain together indicate that the value chain holds great promise and presents several opportunities. These mainly relate to sustainability, since the development and advancement of the chain will contribute to a reduction in the use of natural resources, being an alternative source of protein and a substitute for traditional feed and allowing for the recycling of organic products into compound fertilizers, thus contributing to the Planet, Profit and People model.

The use of insects as animal feed is related to several SDGs, but specifically to SDGs 2 (Zero Hunger and Sustainable Agriculture), 12 (Responsible Consumption and Production) and 13 (Action Against Global Climate Change), in addition to 6 (Drinking Water and Sanitation) and 14 (Life in Water).

**Author Contributions:** Conceptualization, J.G.C.G. and M.T.O.; methodology, J.G.C.G. and E.L.U.; investigation, J.G.C.G.; original draft preparation, J.G.C.G.; writing—review and editing, M.T.O. and H.d.C.L.d.S. All authors have read and agreed to the published version of the manuscript.

**Funding:** This research received in part external funding by the Coordination for the Improvement of Higher Education Personnel—Brazil (CAPES)—Finance Code 001.

**Institutional Review Board Statement:** Not applicable.

**Informed Consent Statement:** Not applicable.

**Data Availability Statement:** Not applicable.

**Acknowledgments:** Special thanks to the companies that participated in the interviews, allowing us to obtain the data used for this study.

**Conflicts of Interest:** The authors declare no conflict of interest.

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
