# Peer review of "Insect Production for Animal Feed: A Multiple Case Study in Brazil"

_sustainability, doi:10.3390/su151411419_

Round 1
Reviewer 1 Report
The manuscript, titled "Insect Production for Animal Feed: A Multiple Case Study in Brazil," is of great interest to readers as the authors delve into the topic of insect production for animal feed. However, it is recommended that the manuscript undergo review by a native English speaker. Additionally, the length of the paragraphs should be increased to provide more comprehensive information.
While the authors place emphasis on the protein content of insects, it is important to discuss the other nutrients present in insects as well. Exploring the nutritional composition of insects would provide a more holistic understanding of their potential as animal feed.
In the introduction section, it would be beneficial to expand on the need for specific nutrients and the varying quantities of feed required for different animals. By addressing these factors, the authors can provide a foundation for the subsequent discussions on insect production for animal feed.
On line 34, it is suggested that the authors mention the importance of reducing meat consumption. Highlighting this point would underscore the significance of alternative protein sources, such as insects, in addressing sustainability and environmental concerns.
Furthermore, on line 51, the authors could provide an overview of the situation in other countries. Comparing and contrasting the state of insect production for animal feed in different regions would offer valuable insights and broaden the scope of the study.
Lastly, on line 146, the authors should include an introductory paragraph explaining that they will be discussing the three Ps: profit, people, and planet. This framework will guide the subsequent analysis and demonstrate the comprehensive nature of their study.
By implementing these corrections and expanding on the relevant sections, the manuscript will be enhanced, ensuring its readability and providing a more comprehensive understanding of the subject matter.
However, it is recommended that the manuscript undergo review by a native English speaker
Author Response
Dear Editors and Referees,
We appreciate the opportunity to submit an article to this prestigious journal.
We tried to answer or justify all the recommendations and suggestions presented and this certainly improved the quality of this article.
We can highlight that proofreading was carried out by MDPI according to the attached certificate, we rewrote the entire article according to the suggestions that can be seen in the increase of the article from 9 pages to 16 pages, from 4000 words to 8000 words and we added 24 new references.
The responses to each referee's suggestions are presented below.
Best regards
Authors
Reviewer 1
Question (Q) 1 - The manuscript, titled "Insect Production for Animal Feed: A Multiple Case Study in Brazil," is of great interest to readers as the authors delve into the topic of insect production for animal feed. However, it is recommended that the manuscript undergo review by a native English speaker. Additionally, the length of the paragraphs should be increased to provide more comprehensive information.
Answer (A) 1- The article passed MDPI proofreading as certified and the paragraphs were rewritten.
Q2 - While the authors place emphasis on the protein content of insects, it is important to discuss the other nutrients present in insects as well. Exploring the nutritional composition of insects would provide a more holistic understanding of their potential as animal feed.
A2 - We add in item 2 (state of the art) the nutritional composition of insects and their contribution to animal feed.
Q 3 - In the introduction section, it would be beneficial to expand on the need for specific nutrients and the varying quantities of feed required for different animals. By addressing these factors, the authors can provide a foundation for the subsequent discussions on insect production for animal feed.
A3 - We add in item 2 - state of the art, the need for specific nutrients and the varying amounts of food needed by different animals.
Q4 - On line 34, it is suggested that the authors mention the importance of reducing meat consumption. Highlighting this point would underscore the significance of alternative protein sources, such as insects, in addressing sustainability and environmental concerns.
A4 - We put a paragraph in the article that reports the importance of reducing meat consumption
Q5 - Furthermore, on line 51, the authors could provide an overview of the situation in other countries. Comparing and contrasting the state of insect production for animal feed in different regions would offer valuable insights and broaden the scope of the study.
A5 - Several paragraphs have been added that provide an overview of the situation in other countries regarding the production of insects for animal feed.
Q6 - Lastly, on line 146, the authors should include an introductory paragraph explaining that they will be discussing the three Ps: profit, people, and planet. This framework will guide the subsequent analysis and demonstrate the comprehensive nature of their study.
A6 - The paragraph was included, discussing the three Ps: profit, people and planet.
By implementing these corrections and expanding on the relevant sections, the manuscript will be enhanced, ensuring its readability and providing a more comprehensive understanding of the subject matter.
A- We are grateful for the suggestions of reviewer 1 and we try to respond to all suggestions and thus improve the article.

Reviewer 2 Report
The manuscript is well written and serves a purpose that is very interesting. The results that have been presented, however, are neither effective nor sufficient. It is my recommendation that the article be dismissed.
1. What is the main question addressed by the research?
-Assessing the feasibility of using insects as sole or primary ingredients in animal feed in an effort to save natural resources
2. Do you consider the topic original or relevant in the field? Does it
address a specific gap in the field?
-Despite the topic's importance to the field of sustainability, the author fails to outline the work's primary goals.
3. What does it add to the subject area compared with other published
material?
-The authors did a thorough analysis of prior literature, so they added nothing new to the discussion.
4. What specific improvements should the authors consider regarding the
methodology? What further controls should be considered?
-To make their work accessible to as many chicken farmers as possible, authors should take into account the most cost-effective means of cultivating these insects.
5. Are the conclusions consistent with the evidence and arguments presented and do they address the main question posed?
-In this case, yes, the conclusions jived with the evidence and arguments offered.
6. Are the references appropriate?
-Yes, it is.
7. Please include any additional comments on the tables and figures.
-It were published works, so nothing can be changed or added.
Author Response
Dear Editors and Referees,
We appreciate the opportunity to submit an article to this prestigious journal.
We tried to answer or justify all the recommendations and suggestions presented and this certainly improved the quality of this article.
We can highlight that proofreading was carried out by MDPI according to the attached certificate, we rewrote the entire article according to the suggestions that can be seen in the increase of the article from 9 pages to 16 pages, from 4000 words to 8000 words and we added 24 new references.
The responses to each referee's suggestions are presented below.
Best regards
Authors
Reviewer 2
Q0- The manuscript is well written and serves a purpose that is very interesting. The results that have been presented, however, are neither effective nor sufficient. It is my recommendation that the article be dismissed.
A0 - To improve the quality of the article, we rewrote the entire article and added more items in introduction, state of the art, results, discussion and conclusion, we increased from 4000 words to 8000 words and 24 references were added.
Q1. What is the main question addressed by the research?
-Assessing the feasibility of using insects as sole or primary ingredients in animal feed in an effort to save natural resources
A1- We rewrite the objective to: This article aims to analyze the value chain of using edible insects in animal feed in Brazil through the framework of SWOT and a sustainable business model canvas and multiple case study, highlighting the sustainable characteristics.
Q2. Do you consider the topic original or relevant in the field? Does it address a specific gap in the field?
-Despite the topic's importance to the field of sustainability, the author fails to outline the work's primary goals.
A2- We rewrite the objective to: This article aims to analyze the value chain of using edible insects in animal feed in Brazil through the framework of SWOT and a sustainable business model canvas and multiple case study, highlighting the sustainable characteristics. And we relate to the objectives of the ODS.
Q3. What does it add to the subject area compared with other published material?
-The authors did a thorough analysis of prior literature, so they added nothing new to the discussion.
A3- We tried to add to the article new 24 references and information on the subject and apply them in the Brazilian chain.
Q4. What specific improvements should the authors consider regarding the methodology? What further controls should be considered?
-To make their work accessible to as many chicken farmers as possible, authors should take into account the most cost-effective means of cultivating these insects.
A4 - The Brazilian chain is new, and we identified only 5 companies, which prevents us from considering the most economical means, but we present the study by Shah et al. [12] on the economic values of insects compared to other protein sources.
Q5. Are the conclusions consistent with the evidence and arguments presented and do they address the main question posed?
-In this case, yes, the conclusions jived with the evidence and arguments offered.
A5 - We added new items in the conclusion as we added to the article such as the use of the swot framework to analyze the chain.
Q6. Are the references appropriate?
-Yes, it is.
A6 - We added 24 new references.
Q7. Please include any additional comments on the tables and figures.
-It was published works, so nothing can be changed or added.
A7- New tables and figures have been added.

Reviewer 3 Report
Dear Author
Comments enclosed

Need extensive editing
Author Response
Dear Editors and Referees,
We appreciate the opportunity to submit an article to this prestigious journal.
We tried to answer or justify all the recommendations and suggestions presented and this certainly improved the quality of this article.
We can highlight that proofreading was carried out by MDPI according to the attached certificate, we rewrote the entire article according to the suggestions that can be seen in the increase of the article from 9 pages to 16 pages, from 4000 words to 8000 words and we added 24 new references.
The responses to each referee's suggestions are presented below.
Best regards
Authors
Reviewer 3
Q1 - Abstract - It has only introduction and materials and methods and no results component. Abstract should contain 2/3 results and no information on number of samples studied and the place of study etc.
A1 - The abstract has been rewritten and focusing more on results.
Q2 – Introduction - Has several small paragraphs and may be merged for better readability.
A2- The small paragraphs were merged, and the introduction was rewritten.
Q3 - Line 41 - In report 638????
A3 - The line has been corrected indicating the author and research of this reference.
Q4 – Currently in Europe there is a supply-demand gap between the actors in the insect chain.
A4- The sentence has been corrected as per suggestion.
Q5 - 55-88 - May nit fit to the article exactly and may be modified with the more detailed research on black soldier flies and utility and practical application of the present study.
A5- Added in the article more information about black soldier flies, research and applications.
Q6 -Materials and methods - No information on samples size and location of collection, what type of statistical analysis used not present.
A6- The sample were two Brazilian companies Alpha and Beta that were used in the case studies. There was no statistical analysis because the research is qualitative.
Q7- Materials and methods have alpha and beta companies and what is the purpose of this study.
A7- Brazilian companies Alpha and Beta were used in the case studies and the objective was rewritten to: This article aims to analyze the value chain of using edible insects in animal feed in Brazil through the framework of SWOT and a sustainable business model canvas and multiple case study, highlighting the sustainable characteristics.
Q8- Results - Is just like a narration and no tables and statistical data to substantiate these findings.
A8- Results It's like a narration and there are no tables and statistics to substantiate these findings. The case studies were used to characterize the companies as well as the canvas and swot frameworks, which is why the results have these narrative and descriptive characteristics. The authors opted for a qualitative analysis because the data obtained with the research instruments did not allow a quantitative analysis, even a discrete statistical one.
Q9- Discussion- Not comprehensive and need more detailed analysis of marker structure and comparison with the other existing market operations is need.
A9- The chain of insects for animal feed is new in most continents and countries, several chains do not exist and are being built, as is the case in Brazil, which is why we added, in item 2 - state of the art, how are some markets in the main continents and countries.
Q10- Conclusion - Too long and not informative to the broader audience
A10- The conclusion was rewritten as changes were made throughout the article.

Round 2
Reviewer 1 Report
Authors made appropriate corrections and manuscript could be published as it is
Reviewer 2 Report
The subject is timely and will be of interest to the journal's readers. Nonetheless, the manuscript is well-written and the ideas are organised logically. The literature review is exhaustive, providing the reader with an adequate background on the topic.
Reviewer 3 Report
Dear Author
the article has been modified and may be considered for publication